# Clinical Evaluation of COVID-19 Survivors at a Public Multidisciplinary Health Clinic

**DOI:** 10.3390/biomedicines13081888

**Published:** 2025-08-03

**Authors:** Ariele Barreto Haagsma, Felipe Giaretta Otto, Maria Leonor Gomes de Sá Vianna, Paula Muller Maingue, Andréa Pires Muller, Nayanne Hevelin dos Santos de Oliveira, Luísa Arcoverde Abbott, Felipe Paes Gomes da Silva, Carolline Konzen Klein, Débora Marques Herzog, Julia Carolina Baldo Fantin Unruh, Lucas Schoeler, Dayane Miyasaki, Jamil Faissal Soni, Rebecca Saray Marchesini Stival, Cristina Pellegrino Baena

**Affiliations:** 1School of Life Sciences, Pontifícia Universidade Católica do Paraná (PUCPR), Curitiba 80215-901, Paraná, Brazil; ariele.barreto-haagsma@newcastle.ac.uk (A.B.H.); felipe.otto@pucpr.edu.br (F.G.O.); maria.vianna@pucpr.br (M.L.G.d.S.V.); paula.muller@pucpr.br (P.M.M.); andrea.muller@pucpr.br (A.P.M.); paes.silva@pucpr.edu.br (F.P.G.d.S.); carolline.klein@pucpr.edu.br (C.K.K.); debora.herzog@pucpr.edu.br (D.M.H.); julia.baldo@pucpr.edu.br (J.C.B.F.U.); lucas.schoeler@pucpr.edu.br (L.S.); dayane.miyasaki@pucpr.br (D.M.); jamil.faissal@pucpr.br (J.F.S.); rebecca.stival@pucpr.br (R.S.M.S.); 2Hospital Universitário Cajuru, Curitiba 80050-350, Paraná, Brazil; nayanne.oliveira@pucpr.edu.br (N.H.d.S.d.O.); luisa.abbott@hospitalcajuru.com.br (L.A.A.); 3Hospital Marcelino Champagnat, Curitiba 80050-370, Paraná, Brazil

**Keywords:** SARS-CoV-2, COVID-19, long COVID, post-acute COVID-19 syndrome, public health

## Abstract

**Background/Objectives:** This study aimed to evaluate sociodemographic factors, features of the acute infection, and post-infection health status in survivors of COVID-19, assessing their association with post-acute COVID-19 syndrome (PACS). **Methods:** A multidisciplinary public clinic in Brazil assessed COVID-19 survivors between June 2020 and February 2022. Patients were classified as having PACS or subacute infection (SI). Data on the history of the acute infection, current symptoms, physical examination, and laboratory findings were collected and analyzed using multivariate models with PACS as the outcome. **Results:** Among the 113 participants, 63.71% were diagnosed with PACS at a median of 130 days (IQR: 53–196) following acute symptom onset. Admission to the intensive care unit was more frequent among individuals with PACS than those with SI (83.3% vs. 65.0% respectively; *p* = 0.037). Symptoms significantly more prevalent in the PACS group when compared to the SI cohort included hair loss (44.4% vs. 17.1% respectively; *p* = 0.004), lower limb paresthesia (34.7% vs. 9.8% respectively; *p* = 0.003), and slow thinking speed (28.2% vs. 0.0% respectively; *p* < 0.001). Logistic regression revealed that only the time interval between the onset of acute symptoms and the clinical evaluation was independently associated with a PACS diagnosis (β = 0.057; 95% CI: 1.03–1.08; *p* < 0.001). **Conclusions:** Patients with PACS had a higher frequency of intensive care unit admission compared to those with subacute infection. However, in the multivariate analysis, the severity of the acute infection did not predict the final diagnosis of PACS, which was associated only with the time elapsed since symptom onset.

## 1. Introduction

In late 1889, the first recorded pandemic caused by the influenza virus began in St. Petersburg and rapidly spread through Europe, becoming known as the Russian Influenza. Although its official end was declared in 1892, survivors continued to report persistent symptoms even three months after the acute onset of the condition [1]. In addition to its parallels with the recent pandemic scenario, the Russian Influenza illustrates that the formal end of a pandemic does not necessarily mark the end of illness in affected individuals. Rather, it may signify the beginning of a hidden pandemic driven by the long-term sequelae of the initial infection.

The rapid spread of COVID-19 resulted in over 772 million confirmed cases globally, with approximately 7 million deaths. In Brazil alone, around 38.7 million cases and more than 711,000 deaths have been documented [2]. Seeking to prevent a repeat of the historical oversight observed during the 1890s pandemic when persistent symptoms remained largely unrecognized, the term post-acute COVID-19 syndrome (PACS) was introduced [3]. The most widely accepted definition, provided by the World Health Organization (WHO), characterizes PACS as new or persistent symptoms occurring at least three months after the acute onset of infection, lasting for at least two months, and not explained by alternative diagnoses in individuals with a confirmed or probable history of COVID-19 [4].

In terms of its clinical features, PACS encompasses a broad and heterogeneous array of symptoms, similar to its acute phase. However, there is no clear consensus regarding symptom attribution: it remains uncertain which manifestations are directly caused by chronic COVID-19, which are residual effects of the subacute infection, and which overlap with conditions such as post-intensive care syndrome [3,5].

While uncertainty remains regarding the precise origins of these diverse symptoms, their prevalence among large numbers of COVID-19 survivors poses an unprecedented challenge. Health systems worldwide, including Brazil’s Unified Health System (SUS), England’s National Health Service (NHS), and the privatized model in the United States, are expected to face significant pressure in scaling up multidisciplinary and long-term care for PACS. Although these systems differ substantially in structure and funding, none were designed to absorb the volume and complexity of post-COVID 19 sequelae, characterized by multisystem involvement, fluctuating symptomatology, and prolonged duration [6,7]. In order to effectively adapt care pathways and manage the complexity of post-COVID 19 sequelae, these systems must first base their strategies on a clear understanding of the needs of the survivors, obtained through well-designed epidemiological studies.

Huang et al. [8] conducted the largest longitudinal study of COVID-19 survivors to date. Their findings demonstrated increased incidence of symptoms such as dyspnea and depression 12 months after discharge, compared to a six-month follow-up. However, the study was limited by selection bias, as most participants had been hospitalized early in the pandemic, and detailed clinical data from the acute phase were lacking, impairing the identification of early predictors of PACS [8].

Similarly, de Oliveira J.F. et al. [9] offered a Brazilian perspective on the epidemiology of PACS. Their multivariate analysis identified dysgeusia during the acute phase, intensive care unit (ICU) admission, and time from symptom onset to follow up as independent risk factors for the development of PACS. However, follow up was conducted solely through telephone interviews, with physical examination findings being self-reported by participants, limiting clinical accuracy.

While numerous epidemiological studies have described COVID-19 sequelae [10,11,12,13,14,15], few have been conducted in low- and middle-income settings [9,16,17,18,19], and fewer still include comprehensive data from acute, hospitalization, and chronic phases, including physical examination, functional testing, and laboratory evaluation. Furthermore, the increased economic and social burden caused by the frequent and persistent symptoms among COVID-19 survivors highlight an urgent need to understand the risk factors and patterns of post-COVID 19 sequelae to develop sustainable care models for long-term COVID-19 recovery. In an effort to address such gaps, the present study explores sequelae patterns in a cohort of adult Brazilian COVID-19 survivors referred to a public multidisciplinary clinic, with the objective of outlining the sociodemographic and clinical profile of COVID-19 survivors in a developing country and identifying the prevalence of PACS. This study also hypothesizes that specific sociodemographic and clinical characteristics, along with features of the acute infection, could be utilized as possible predictors of the final diagnosis of PACS.

## 2. Materials and Methods

### 2.1. Population and Study Design

A prospective and observational cohort study was conducted with survivors of COVID-19 acute infection discharged from hospitals and directly referred by the basic healthcare units of the state of Paraná, Brazil. They were then referred to our public multidisciplinary clinic in Curitiba, Paraná, Brazil, between June 2020 and February 2022, where a follow-up consultation with a respiratory clinician and physiotherapist was completed. All patients were referred for routine blood tests and were contacted by telephone at least 3 months after the first follow up as well.

Inclusion criteria required individuals to be over 18 years old by the time of the consultation, with a history of acute COVID-19 infection proved by a positive reverse transcriptase–polymerase chain reaction (RT-PCR) test or rapid antigen test for SARS-CoV-2. Patients without data about their sociodemographic profile, acute infection history, or electronic report were excluded from the study.

The entire study population presented with at least one persistent symptom after the acute phase of COVID-19; thus, patients were diagnosed with either post-acute COVID-19 syndrome (PACS) or subacute infection (SI) according to the definitions suggested by the World Health Organization. PACS was defined as symptoms or signs that occur in individuals after 12 weeks of the start of the probable or confirmed acute infection by COVID-19 and that remain for at least 8 weeks and cannot be explained by an alternative diagnosis. Meanwhile, SI was defined as symptoms or signs that start or persist after 4 weeks of the beginning of the acute infection by COVID-19, remaining for a maximum of 12 weeks [4].

The Local Ethics Committee approved the study protocol (protocol CAAE 30188020.7.1001.0020).

### 2.2. Variables

#### 2.2.1. Baseline Variables

These included baseline demographics (age, sex, smoking and drinking status), type and number of comorbidities, COVID-19 vaccination status, number of previous hospital admissions, number of previous surgeries, clinical data about the acute period of COVID-19 infection (number of symptoms, types of symptoms, close contact contamination and deaths, need for hospitalization) and were all questioned directly to the patient at the time of the follow up. Clinical data about the hospitalization period included days from onset of symptoms to hospitalization, length of hospitalization, use, type, and length of supplementary oxygenation, complications, ICU admission, and length of stay. The latter were collected directly from patient reporting or through medical records brought by the patients.

#### 2.2.2. Follow-Up Variables

During the consultation, the pulmonary clinician measured anthropometric measurements such as the body mass index (BMI), and abdominal and cervical circumference, and performed the modified Medical Research Council scale for dyspnea (mMRC) [20]; a complete multisystemic physical examination; and the mini mental state examination [21].

On the other hand, with the physiotherapist, patients were evaluated with a pulmonary function physical examination, strength test [22], handgrip strength measurements with a dynamometer bilaterally (model SAEHAN-SH5001, Carci^®®^, São Paulo, Brazil), exercise testing (6 min walking test (6MWT) [23], manovacuometry measurements (model M120, Comercial Médica^®®^, Imbiribeira, Recife, Brazil), and peak flow tests with a Peak Flow Meter (Physical Care^®®^, São Paulo, Brazil). All tests and procedures were conducted using validated guidelines [22,24,25,26,27].

The laboratory variables included serum glucose (mg/dL); serum uric acid (mg/dL); creatinine (mg/dL); urea (mg/dL); total and fractions of bilirubin (mg/dL); aspartate and alanine aminotransferases (UI/L); alkaline phosphatase (UI/L); gamma- glutamyl tansferase (UI/L); sodium (mEq/L); potassium (mEq/L); full blood count; fibrinogen (mg/dL); glycated hemoglobin (%); thyroid stimulant hormone (microunits/mL); albumin (g/dL); ferritin (ng/mL); C-reactive protein (mg/dL); total IgE (kilounits/L); D-dimers (nanograms/L); troponin (pg/mL); and a lipidogram.

Patients also received a telephone call after a minimum of 3 months after the first follow up in which their quality of life and symptoms were questioned; however, the results of these telephone calls are not yet included in this study.

### 2.3. Statistical Analysis

All statistical analyses were performed using the Statistical Package for the Social Science (SPSS, IBM Corp. Released 2020, Version 28.0, Armonk, NY, USA), and statistical significance was inferred at *p* < 0.05. All the covariates were stratified by diagnosis of PACS or SI, and missing data were imputed where applicable. Continuous variables with a normal distribution are presented as the means with their respective standard deviation (SD) and analyzed with Student’s *t*-test, while continuous variables with a not-normal distribution were described through their median and interquartile range (IQR), and analyzed with the Mann–Whitney test. Categorical variables are reported as frequencies with their respective percentages, and were tested with the Chi-square or Fisher tests.

Variables from the univariate analysis with *p* < 0.2 were sorted into binary logistic regression models in which diagnosis of PACS was the dependent variable. The Wald test was used to evaluate the significance of each covariate in the models, and the values were described as β coefficients, odds ratio (OR), and 95% confidence intervals (95% CI).

Initially, all the analyses were adjusted only for the time from onset of acute symptoms and follow up (Model 1), then for sex (Model 2), and then for the number of previous hospitalizations and comorbidities (Model 3). Other adjustments were made for vaccine status (only considered completed if two doses were received) (Model 4), and finally for the length of hospital stay (Model 5).

## 3. Results

### 3.1. Baseline Characteristics

From the final overall sample of 113 participants included in the study, 53.1% were female (subacute infection [SI] 43.9%, post-acute COVID-19 syndrome [PACS] 58.3%). The baseline characteristics are presented in Table 1. The mean age was distributed equally between groups, being 52.9 ± 14.6 years in the SI group and 51.10 ± 13.0 years in the PACS group.

Most of the overall population had at least one comorbidity (81.4%); this was more prevalent in the SI group than in the individuals with PACS (SI 92%, PACS 75.0%, *p* = 0.024). The most frequent comorbidities in the overall cohort were arterial hypertension (61.1%), dyslipidemia (31.0%), and diabetes (25.7%). The overall and group-specific distribution of the comorbidities is presented in Figure 1**,** but a more detailed description can be found in Table 1. It was observed that 69.4% of the population was already fully vaccinated at the time of consultation and that individuals diagnosed with PACS had significantly higher proportions of the first (83.3%) and second dose of vaccination (73.3%) (*p* = 0.030 and 0.023, respectively).

### 3.2. Acute COVID-19 Characteristics

There was no difference in the median number of symptoms during the acute phase of infection between groups, with a median of 4 (IQR 0–12) symptoms in the overall study population. A more detailed description of the infection’s acute phase is portrayed in Table 2. No acute symptoms were significantly related to the final PACS diagnosis, and the most frequent ones among the overall population were dyspnea (74.3%), fatigue (55.8%), fever (50.4%), and myalgia (46.9%).

Most of the individuals of our sample (98.3%) were hospitalized due to acute COVID-19 infection, and the median time from onset of acute symptoms to hospitalization was the same throughout the two groups (8 days). Individuals diagnosed with PACS presented with a longer length of stay in the hospital (median 14.5 [IQR 7.0–25.2] days) when compared to those with SI (median 9.0 [IQR 6.0–19.5] days), but there was no significant difference in the comparison. However, a significantly higher proportion of patients admitted into the ICU was reported in the PACS group (SI 65.0%, PACS 83.3%, *p* = 0.037). Furthermore, the PACS group presented with a significantly higher proportion of patients with the use of non-invasive mechanical ventilation and nasal cannula (*p* = 0.026 and *p* = 0.022, respectively).

### 3.3. Post-COVID-19 Characteristics

The median time between the onset of acute symptoms and follow up was 130 days (IQR 53–196), and 72 patients (63.71%) were diagnosed with post-acute COVID-19 syndrome. In the overall sample, the median number of sequelae reported was 9 (IQR 1–31). The most prevalent persistent symptoms were weight loss (70.8%), dyspnea (67.3%), and fatigue (57.5%) (Figure 2).

The sequelae that were observed with a significant higher prevalence in the group diagnosed with SI were (% SI; % PACS; p) weight loss (85.4%; 52.5%; 0.011), dyspnea (80.5%; 59.7%; 0.036), cough (48.8%; 19.4%; 0.001), and depression and/or anxiety (14.6%; 2.8%; 0.049). Among the persistent symptoms that were significantly more prevalent in between the individuals diagnosed with PACS were the following: hair loss (17.1%; 44.4%; 0.004); lower limb paranesthesia (9.8%; 34.7%; 0.003); slow thinking speed (0.0%; 28.2%; <0.001); rhinorrhea (2.4%; 22.2%; 0.005); and anosmia (4.9%; 19.4%; 0.047). In order to better visualize which category of symptoms where present in the two groups, the difference in between symptoms’ prevalence was calculated, and symptoms were grouped together by bodily systems (Figure 3).

A summary of the physical examination findings is provided in Table 3 to facilitate a clearer understanding of the patient’s clinical presentation, but a more detailed description can be found in Appendix A. The heart rate was found to be significantly different between the groups (*p* = 0.010), with a mean of 87.4 ± 15.4 bpm in individuals with SI and 80.0 ± 13.8 bpm in patients with PACS. An elevated respiratory rate was found in 50.0% of the patients with SI and in only 28.2% of those with PACS (*p* = 0.035).

There was no statistical significance, but it is interesting to note that the study population’s abdominal circumference was higher (mean 103.8 ± 12.9 cm) than the recommended levels by the WHO [28]. In addition, more than 60% of the overall population was classified with a body mass index (BMI) ≥ 30 kg/m^2^ (overall 62.6%; SI 65.8%; PACS 60.9%). Furthermore, the overall sample and each group presented median values of glycated hemoglobin above normalcy, being considered pre-diabetic already (median 5.7 [IQR 5.4–6.2] %). The complete laboratory findings of our study can be found in Appendix A.

The complete list of variables and tests carried out in the population by the physiotherapist is presented in Appendix A. The % of predicted distance walked was significantly lower in the PACS sample, which walked a mean of only 71.9 ± 18.1% of the predicted distance for the test, while individuals with SI walked a mean of 89.9 ± 20.2% (*p* = 0.003) (Table 2). In a more detailed analysis, it was observed that such difference remained only among men. Men with SI walked a mean of 101.1 ± 16.8% of the predicted walking distance, while the value for those diagnosed with PACS was of 74.7 ± 13.4% (*p* < 0.001).

### 3.4. Factors Associated with Post-Acute COVID-19 Syndrome

In the multivariate analysis, a binary logistic regression was used with the diagnosis of PACS as the dependent variable. All the five models can be found in Appendix A. In all the proposed models, the time between acute onset and supplementary follow up was the only variable independently and positively associated with the diagnosis of PACS (OR 1.058, IC 95% (1.033–1.084), *p* < 0.001) (Figure 4).

## 4. Discussion

Our results show higher rates of post-acute COVID-19 syndrome even months after the onset of acute symptoms when compared to a recent meta-analysis of 194 studies (63% in our study and 45% in the meta-analysis) [17]. The high rates of post-acute COVID-19 syndrome (PACS) prove that the control of the acute pandemic is not synonymous to the end of disease, which will most likely impact healthcare systems worldwide due to high demand from the increasing number of chronically ill patients, reaching now more than 625 million survivors internationally [2].

The *CO-FRAIL* study [18], also carried out in Brazil, found a rate of 56% of PACS within a population of ≥ 50 years old individuals that were previously hospitalized. However, rates as high as 87.4% have already been documented by the Brazilian and international literature [9]. That being said, we believe the true prevalence of PACS within our population might be even higher due to two main reasons, though a longer follow-up time would be needed to fulfil the WHO diagnosis criteria of PACS [4].

Firstly, our sample was made up mostly of patients with a history of severe acute infection: 93.8% of our overall population was hospitalized and 76.4% required intensive care, much higher proportions when compared to other studies [18,19,29]. Arnold et al. [11] found that nearly three-quarters of their previously hospitalized patients remained with symptoms after 3 months. Another study showed a PACS incidence of 83.5% 6 months after discharge from intensive care unit (ICU) [30]. The second reason is the scarcity of rehabilitation programs for COVID-19 survivors in a developing country, which results in high demand. This hypothesis might both underestimate and overestimate the real prevalence of PACS because it narrows our population sample, thus overestimating the values, but it can also deprive more mild cases of PACS from receiving care, thus underestimating the rates of diagnosis.

One point of interest that has not been tackled in many studies is vaccination status. Miranda et al. [19] presented a rate of only 0.8% of fully vaccinated patients. Our study presented with 50.4% of individuals already fully vaccinated by the time of consultation. It was also observed that there was a significant difference of vaccination status between the groups: while 69.4% of the PACS groups was already fully vaccinated, only 17.1% of the individuals with subacute infection (SI) were (*p* < 0.001). In this study, the date of vaccination was not compared to the date of infection, which could explain the difference found in this analysis. Furthermore, recent evidence indicates that prior vaccination reduces the risk of PACS and that additional doses confer incremental protection [31].

Our study was able to clearly showcase that there is a significant difference between the persistent symptoms presenting in subacute COVID-19 and PACS. Individuals with PACS had significantly higher percentages of specific sequelae such as hair loss, lower limb paranesthesia, slow thinking speed, rhinorrhea, and anosmia. The symptoms of PACS are greatly diverse in the literature, affecting many different systems such as the respiratory, cardiovascular, musculoskeletal, cutaneous, and neurological systems, but are at similar rates compared to our sample [6,32].

Albeit no statistical significance in the intergroup comparison, it is interesting to note that the overall body mass index (BMI) and abdominal circumference values found in our sample were above recommendations by the World Health Organization (WHO) [28]. The same is true regarding the median of glycated hemoglobin, found to be 5.7% in our sample, a value already considered to be pre-diabetic [33]. These observations reveal the profile of the acutely severe population, reinforcing that obesity predisposes an unfavorable clinical course of the disease [3,34].

Exercise testing, such as the 6 min walking test, is widely used in PACS-related studies to analyze pulmonary function [6,35]. In our study, it was found that not only the PACS population walk significantly less of what was expected (*p* = 0.003) but that, when classified by sex, this observation maintained itself only in the male population (*p* < 0.001). Shah et al. [36] observed a mean percentage of predicted distance walked of 96%, higher than what was found in our study (71.9%).

The only risk factor for developing PACS in our study was the time from acute onset to follow up (OR 1.058, IC 95% 1.01–1.07, *p* = 0.003). In a longitudinal prospective study of one year after hospital discharge, it was found that female sex, BMI, previous history of ICU admission, and hospitalization length were all positively related to the diagnosis of PACS in individuals over 50 years old [18]. The development of PACS was also associated with dysgeusia and time from acute onset in a study including 439 previously hospitalized patients [9]. After a follow-up period of 6 months of previously hospitalized patients due to COVID-19, it was found that the number of symptoms and the severity of acute infection were both associated with post-COVID-19 syndrome [37]. However, all studies previously mentioned, albeit having a sample only of previously hospitalized patients, included 61%, 19%, and 0% of overall ICU admissions, respectively, while in our study, the ICU admission rate was 76.4% in the overall sample and 83.3% in the PACS group. There are two main hypotheses for our results.

The first hypothesis includes the discovery that PACS is independent of any risk factors in a population similar to ours, with a history of acute COVID-19 infection of high severity, and that the time from onset is enough to reveal COVID-19 sequelae. However, there is insufficient evidence in the literature to support this; nevertheless, the very existence of such hypothesis underscores the pressing need for post COVID-19 care. The second hypothesis is the masking of PACS diagnosis by the symptoms of post-intensive care syndrome (PICS) [5,6,9,38]. The greater severity of the acute phase of our sample might explain the lack of association between common risk factors and the development of PACS, but as time goes by, the symptoms of PICS might be attenuated and even treated, leaving only pure post-acute COVID-19 syndrome. Studies with control groups without previous COVID-19 infection are needed to prove this theory [39,40].

### Limitations

The single-center design limits the characteristics of our cohort, and the relatively small sample size restricted statistical analysis to a simpler logistic regression model. Moreover, some variables were self-reported by patients and are, consequently, subjective in nature. While the specific profile of our population may underestimate the association of risk factors related to PACS, it can also shed light on future studies, allowing them to plan study designs to overcome the overlap of syndromes.

## 5. Conclusions

In a population of COVID-19 survivors that is mostly made up of previously hospitalized patients, a high prevalence of PACS was found. However, similar clinical features with post-intensive care syndrome and the absence of other predictors for PACS, aside from time from acute onset, do not allow this study to rule out the possible overlap between PACS and post-intensive care syndrome in the population, which demonstrates the need for more studies with control groups and a better definition of post-acute COVID-19 syndrome. Outlining the differences between these syndromes in previously hospitalized patients will aid in policy-planning for their management.

Moreover, we were able to describe a select group of survivors of severe COVID-19 treated in the public health system of Brazil, identifying the most common sequelae in a multidisciplinary manner, and describing the unified and consistent care offered by a single center.

## Figures and Tables

**Figure 1 biomedicines-13-01888-f001:**
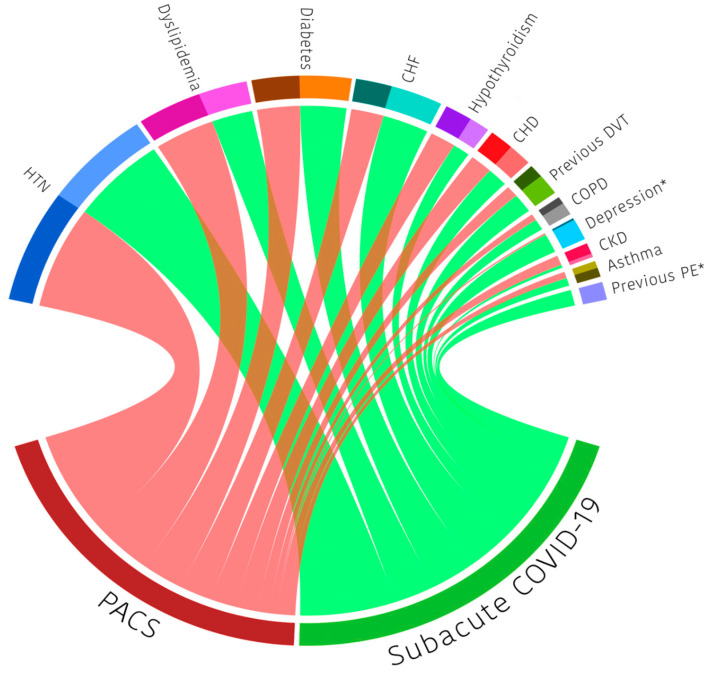
Distribution of comorbidities in the follow up of COVID-19 survivors according to their final diagnosis (*n* = 113). PACS: post-acute COVID-19 syndrome; HTN: hypertension; CHF: congestive heart failure; CHD: chronic heart disease; DVT: deep venous thrombosis; COPD: chronic obstructive pulmonary disease; CKD: chronic kidney disease; PE: pulmonary embolism. All correlations between comorbidities and diagnosis of PACS presented *p* > 0.05, except depression* (*p* = 0.023) and previous PE* (*p* = 0.016). P-value found with Student’s *t*-test for independent samples, the Fisher test, Chi-square test, or Mann–Whitney test according to the classification and distributions of each variable.

**Figure 2 biomedicines-13-01888-f002:**
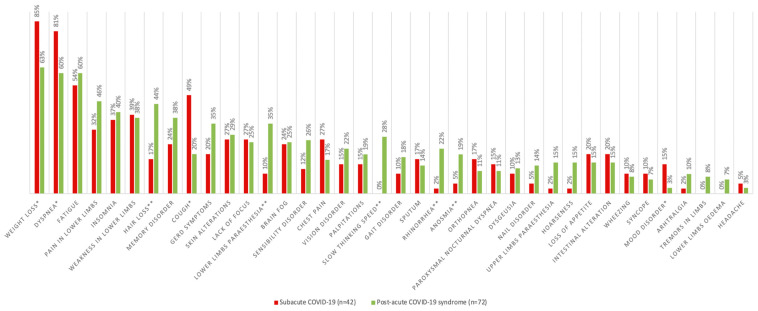
Persistent symptoms in the follow up of COVID-19 survivors according to their final diagnosis (*n* = 113). GERD: gastroesophageal reflux disease. All the correlations between persistent symptoms and diagnosis of PACS presented *p* > 0.05, except weight loss* (*p* = 0.011), dyspnea* (*p* = 0.036), cough* (*p* = 0.001), mood disorder* (*p* = 0.049), hair loss** (*p* = 0.004), lower limb paranesthesia** (*p* = 0.003), slow thinking speed** (*p* < 0.001), rhinorrhea** (*p* = 0.005), and anosmia** (*p* = 0.047). *p*-value found with the Student *t*-test for independent samples, Fisher test, Chi-square test, or Mann–Whitney test according to the classification and distributions of each variable.

**Figure 3 biomedicines-13-01888-f003:**
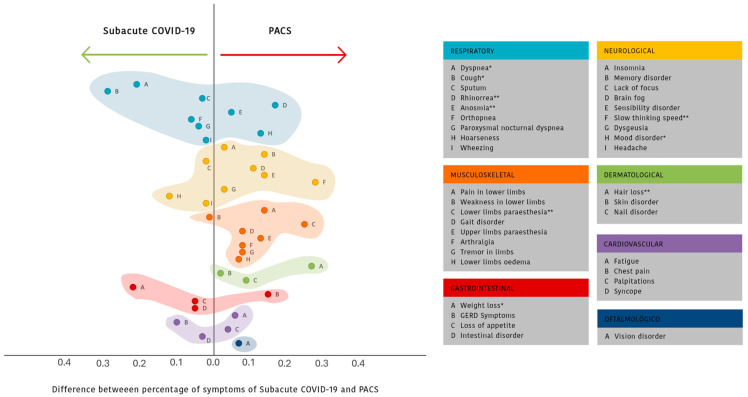
Distribution of symptoms in COVID-19 survivors that were diagnosed either with subacute COVID-19 or PACS during follow up. GERD: gastroesophageal reflux disease. All the correlations between persistent symptoms and diagnosis of PACS presented *p* > 0.05, except weight loss* (*p* = 0.011), dyspnea* (*p* = 0.036), cough* (*p* = 0.001), mood disorder* (*p* = 0.049), hair loss** (*p* = 0.004), lower limb paranesthesia** (*p* = 0.003), slow thinking speed** (*p* < 0.001), rhinorrhea** (*p* = 0.005), and anosmia** (*p* = 0.047). P-value found with the Student *t*-test for independent samples, Fisher test, Chi-square test, or Mann–Whitney test according to the classification and distributions of each variable.

**Figure 4 biomedicines-13-01888-f004:**
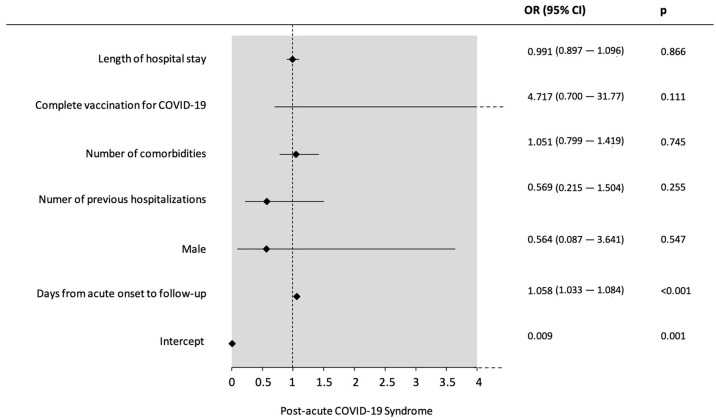
Multivariate analysis for risk factors possibly associated with the diagnosis of post-acute COVID-19 syndrome (Model 5). OR: odds ratio. CI 95%: 95% confidence intervals. *p*-values found with binary logistic regression with the Wald test.

**Table 1 biomedicines-13-01888-t001:** Baseline sociodemographic characteristics of COVID-19 survivors during their follow-up face-to-face consultation according to their final diagnosis (*n* = 113).

	Total (*n* = 113)	Subacute COVID-19 (*n* = 41)	Post-Acute COVID-19 Syndrome (*n* = 72)	*p* *
**Age (years), mean ± SD**	51.7 ± 13.2	52.9 ± 13.6	51.1 ± 13.0	0.484
**Sex, *n* (%)**				0.171
Male	53 (46.9)	23 (56.1)	30 (41.7)	
Female	60 (53.1)	18 (43.9)	42 (58.3)	
**Smoking status, *n* (%)**				0.891
Never smoked	76 (67.3)	28 (68.3)	48 (66.7)	
Smoker	4 (3.5)	1 (2.4)	3 (4.2)	
Past smoker	33 (29.2)	12 (29.3)	21 (29.2)	
**Alcohol consumption, *n* (%)**				0.256
Never drank alcohol	86 (76.1)	30 (73.2)	56 (76.1)	
Occasionally drinks alcohol	24 (21.2)	11 (26.8)	13 (18.1)	
Frequently drinks alcohol	3 (2.7)	0 (0.0)	3 (4.2)	
**Allergies, *n* (%)**	11 (9.7)	4 (9.8)	7 (9.7)	1.000
**Presence of at least one comorbidity, *n* (%)**	92 (81.4)	38 (92.7)	54 (75.0)	**0.024**
**Number of comorbidities, median (IQR)**	2.0 (1.0–4.0)	3.0 (1.5–4.5)	2.0 (0.2–4.0)	0.156
**Comorbidities, *n* (%)**				
Arterial hypertension	69 (61.1)	23 (56.1)	46 (63.9)	0.430
Dyslipidemia	35 (31.0)	10 (24.4)	25 (34.7)	0.295
Diabetes	29 (25.7)	11 (26.8)	18 (25.0)	0.827
Congestive heart failure	25 (22.1)	11 (26.8)	14 (19.4)	0.480
Hypothyroidism	15 (13.3)	4 (9.8)	11 (15.3)	0.299
Coronary heart disease	14 (12.4)	5 (12.2)	9 (12.5)	1.000
Previous deep vein thrombosis	10 (8.8)	5 (12.2)	5 (6.9)	0.492
Peripheral venous insufficiency	6 (5.4)	3 (7.5)	3 (4.2)	0.665
Arrhythmia	6 (5.3)	3 (7.3)	3 (4.2)	0.666
Chronic obstructive pulmonary disease	6 (5.3)	3 (7.3)	3 (4.2)	0.666
Depression	6 (5.3)	5 (12.2)	1 (1.4)	**0.023**
Malignancy	5 (4.4)	2 (4.9)	3 (4.2)	1.000
Chronic renal failure	5 (4.4)	1 (2.4)	4 (5.6)	0.662
Rhinitis	5 (4.4)	2 (4.9)	3 (4.2)	1.000
Asthma	5 (4.4)	2 (4.9)	3 (4.2)	1.000
Previous pulmonary embolism	4 (3.5)	4 (9.8)	0 (0.0)	**0.016**
Peripheral arterial disease	3 (2.7)	0 (0.0)	3 (4.2)	0.552
Cerebrovascular disease	3 (2.7)	1 (2.5)	2 (2.8)	1.000
Autoimmune disease	2 (1.8)	1 (2.4)	1 (1.4)	1.000
Anemia	1 (0.9)	0 (0.0)	1 (1.4)	1.000
Dementia	1 (0.9)	0 (0.0)	1 (1.4)	1.000
**Number of regular use medications, median (IQR)**	1.0 (0.0–4.0)	2.0 (0.0–4.0)	1.0 (0.0–5.0)	0.925
**Previous hospitalization, *n (%)***	55 (48.7)	24 (58.5)	31 (43.1)	0.123
**Number of previous hospitalizations,** **median (IQR)**	0.0 (0.0–1.0)	1.0 (0.0–2.0)	0.0 (0.0–1.0)	0.172
**Previous surgery, *n* (%)**	64 (56.6)	25 (61.0)	39 (54.2)	0.556
**Number of previous surgeries, median (IQR)**	1.0 (0.0–2.0)	1.0 (0.0–2.0)	1.0 (0.0–2.0)	0.617
**First dose of COVID-19 vaccination, *n (%)***	57 (78.1)	7 (53.8)	50 (83.3)	**0.030**
**Second dose of COVID-19 vaccination, *n (%)***	49 (67.1)	5 (38.5)	44 (73.3)	**0.023**
**Close contact contamination with COVID-19, *n (%)***	60 (67.4)	16 (61.5)	44 (69.8)	0.465
**Number of close contacts contaminated,** **median (IQR)**	1.0 (0.0–2.0)	1.0 (0.0–2.0)	1.0 (0.0–3.0)	0.122
**Number of deaths of close contacts, median (IQR)**	0.0 (0.0–0.0)	0.0 (0.0–0.2)	0.0 (0.0–0.0)	0.176

SD: standard deviation; IQR: interquartile range. * *p*-values were found through the Student *t*-test for independent samples, Chi-square test, or Mann–Whitney test according to variable type and distribution.

**Table 2 biomedicines-13-01888-t002:** Characteristics of acute infection and hospitalization due to COVID-19 infection.

	Total (*n* = 113)	Subacute COVID-19 (*n* = 41)	Post-Acute COVID-19 Syndrome (*n* = 72)	** p*
**Number of symptoms during acute infection, *median (IQR)***	4.0 (0.0–12.0)	4.0 (1.0–9.0)	4.0 (0.0–12.0)	0.442
**Signs and symptoms during acute infection, *n (%)***				
Dyspnea	84 (74.3)	29 (70.7)	55 (76.4)	0.511
Fatigue	63 (55.8)	23 (56.1)	40 (55.6)	1.000
Fever	57 (50.4)	20 (48.8)	37 (51.4)	0.846
Myalgia	53 (46.9)	19 (46.3)	34 (47.2)	1.000
Cough	50 (44.2)	14 (34.1)	36 (50.0)	0.118
Anosmia	35 (31.0)	15 (36.6)	20 (27.8)	0.399
Headache	32 (28.3)	7 (17.1)	25 (34.7)	0.053
Dysgeusia	21 (18.6)	8 (19.5)	13 (18.1)	1.000
Diarrhea	15 (13.3)	6 (14.6)	9 (12.5)	0.778
Nausea and vomits	14 (12.4)	4 (9.8)	10 (13.9)	0.767
Shivers	10 (8.8)	3 (7.3)	7 (9.7)	0.745
Sputum	8 (7.1)	1 (2.4)	7 (9.7)	0.255
Excessive sweating	4 (3.5)	1 (2.4)	3 (4.2)	1.000
**Hospitalization, *n* (%)**	106 (93.8)	40 (97.6)	66 (91.7)	0.419
**Hospitalization characteristics**				
Time from acute onset to hospitalization (days), median (IQR)	8.0 (–22.0–20.0)	8.0 (–22.0–20.0)	8.0 (0.0–19.0)	0.922
Length of hospitalization (days), median (IQR)	13.0 (0.0–62.0)	9.0 (0.0–42.0)	14.5 (0.0–62.0)	0.062
ICU admission, *n* (%)	81 (76.4)	26 (65.0)	55 (83.3)	0.037
Length of ICU admission (days), median (IQR)	9.0 (1.0–60.0)	8.0 (1.0–25.0)	9.0 (1.0–60.0)	0.426
Use of nasal cannula, *n* (%)	84 (80.0)	27 (67.5)	57 (87.7)	0.022
Use of non-invasive mechanical ventilation, *n* (%)	65 (61.3)	19 (47.5)	46 (69.7)	0.026
Length of non-invasive ventilation (days), median (IQR)	4.0 (1.0–14.0)	1.0 (1.0–3.0)	7.5 (1.0–14.0)	0.095
Orotracheal intubation, *n* (%)	24 (22.6)	7 (17.5)	17 (25.8)	0.351
Length of orotracheal intubation (days), mean ± SD	14.7 ± 10.9	12.5 ± 5.2	15.7 ± 12.8	0.638
Number of complications during hospitalization, median (IQR)	0.0 (0.0–3.0)	0.0 (0.0–3.0)	0.0 (0.0–2.0)	0.653
Pulmonary embolism during hospitalization, *n* (%)	5 (4.7)	3 (7.5)	2 (3.1)	0.363
Deep vein thrombosis during hospitalization, *n* (%)	5 (4.7)	2 (5.0)	3 (4.5)	1.000
Readmission into hospital after discharge, *n* (%)	3 (2.9)	1 (2.6)	2 (3.2)	1.000

SD: standard deviation; IQR: interquartile range; ICU: intensive care unit; * *p*-values were found through the Student *t*-test for independent samples, Chi-square test, or Mann–Whitney test according to variable type and distribution.

**Table 3 biomedicines-13-01888-t003:** Clinical and anthropometric characteristics of patients with a history of COVID-19, stratified by the phase of the disease.

	Total (*n* = 113)	Subacute COVID-19 (*n* = 41)	Post-Acute COVID-19 Syndrome (*n* = 72)	*p* *
**BMI, median (IQR)**	31.3 (16.7–47.2)	31.6 (16.7–37.7)	31.2 (18.8–47.2)	0.703
**BMI categories, *n* (%)**				0.361
Underweight or normal	10/107 (9.3)	5/38 (13.2)	5/69 (7.2)	
Overweight	30/107 (28.0)	8/38 (21.1)	22/69 (31.9)	
Obese	67/107 (62.6)	25/38 (65.8)	42/69 (60.9)	
**Abdominal circumference (cm), mean** ± **SD**	103.8 ± 12.9	102.5 ± 14.9	104.1 ± 2.5	0.677
**Altered abdominal circumference according to sex, *n* (%)**	55/83 (66.3)	10/14 (71.4)	45/69 (62.5)	0.764
**Heart rate (bpm), mean** ± SD	82.7 ± 14.7	87.4 ± 15.4	80.0 ± 13.8	**0.010**
**Elevated respiratory rate, *n* (%)**	39 (35.8)	19 (50.0)	20 (28.2)	**0.035**
**Oxygen saturation at rest (%), median (IQR)**	96.0 (84–100)	96.0 (85–100)	96 (84–99)	0.132
**Elevated blood pressure, *n* (%)**	50 (45.0)	15 (38.5)	35 (48.6)	0.325
**6-min walking test**				
Predicted walking distance (PWD) (meters), mean ± SD	490.5 ± 39.6	487.4 ± 41.1	492.6 ± 39.1	0.635
Distance walked (meters), mean ± SD	393.8 ± 99.3	444.0 ± 86.0	357.2 ± 95.1	**0.005**
% of PWD, mean ± SD	80.3 ± 21.0	89.9 ± 20.2	71.9 ± 18.1	**0.003**

SD: standard deviation; IQR: interquartile range; BMI: body mass index; * *p*-values obtained through the Student *t*-test, Fisher test, Chi-Square test, or Mann–Whitney test according to the type and distribution of each variable.

## Data Availability

The original contributions presented in this study are included in the article/Appendix A. Further inquiries can be directed to the corresponding author.

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
