# Peer review of "Clinical Evaluation of COVID-19 Survivors at a Public Multidisciplinary Health Clinic"

_biomedicines, 2025, doi:10.3390/biomedicines13081888_

Round 1
Reviewer 1 Report
Comments and Suggestions for Authors
The work is interesting in general. The authors tried to compare the clinical signs and course of PACS and SI. This attempt is of some interest, although it does not fully reveal the significance of PACS in the late period of convalescence. As in other publications, the authors focused on the relatively early period of convalescence (within 6 months). Meanwhile, the symptoms of PACS do not subside during this period. Certain symptoms persist for a longer period. In this regard, I would like to see the features of PACS 12-24 months after recovery. It would be much more interesting to compare the obtained data with the results of examination of healthy volunteers, it is a pity that the authors could not present this.
Reviewer 2 Report
Comments and Suggestions for Authors
This paper describe a retrospective study of patients with acute post-covid 19 syndrome, attempting to obtain correlations between post-covid 19 status and the clinical and socio-demographic profile of the patients.
The article content is clear, relevant for the Public health field and it presented in a well structured manner.
The methods is well described.
The discussion are well structured.
The figures and the tables are easy to interpret and and show well the data.
The conclusions are consistent, based on results of the study.
In my opinion can be published in present form.
Reviewer 3 Report
Comments and Suggestions for Authors
Congratulations for your laborious efforts performed and for the interesting data you presented.
The scientific community is seeking constantly for valuable information that will develop a more comprehensive approach to medical problems.
However, I have some minor comments for your manuscripts aiming to assist you in preparing a more complete presentation of your data.
Comment1: In the abstract body the first sentence in the conclusions does not rely on your results presented. You have either to rephrase the conclusions or to add the missing results that will support your conclusions.
Comment2: It is useful to the reader to describe briefly the study population and the reasoning behind the methodology design to prove that your hypothesis was strong enough to proceed with the study.
Comment3: Some demographics are characterized with widespread therefore the median values should be better used while age for example is related to the frailty syndrome or at least to have statistical normalizations to this effect and similarly to other relevant variables as you already have performed.
Comment4: I am wondering whether the term laboratory evaluation in the title mirrors the content of your discussion and conclusions section while there are so many missing data. Thus, I propose the title be rephrased accordingly.
• Do you consider the topic original or relevant to the field? • What specific improvements should the authors consider regarding the
methodology?
Long Covid as well as the Post Acute syndromes are still considered medically important features while they pose health perils to the patients that also hide a burden to social life, employment and health cost. We then must be supported with clear evidence-based information to make future decisions. We are also aware that the later types of the virus do not provoke the investigated syndromes in the degree that initial types of SARS CoV2 exerted. Therefore, it would be wise to divide the population according to the viral subtype and if possible, to have a control group of patients that following the acute phase they recovered without any objective clinical sign or laboratory abnormal parameter to remain. Even more, you should perform statistical evaluation in the subgroups of the study population who carry hematological and biochemical parameters because these values in regression models may reveal interesting information. As I stated before there is a wide range in the age of the participants and further to the need of median values and to the statistical normalizations it would be necessary to divide the volunteers in age groups because the degree of morbidity varies according to age and when you mix younger and older populations a significant indicative factor will be lost because of statistical correction. In that way then the results will be more beneficial, and your efforts will probably offer more knowledge to fill in the gap
• What does it add to the subject area compared with other published
material?
The possible strength of your study will be firstly the probability to have the post-acute Covid syndrome unmasked while we would experience significant clinical milestones in the acute and the recovery phase that could be indicative of what to expect in an individualized patient approach to assist the patient and provide adequate support. The information in the field is not complete yet and many other studies like yours will be of great benefit. Even more, the limitation that a single study medical center may be another strength because the volunteers all received the same level of health provision and are of similar genetic background.
• Are the conclusions consistent with the evidence and arguments presented and do they address the main question posed? Please also explain why this is/is not the case. What is the main question addressed by the research?
In order for the conclusions to be consistent with your results and to be presented as beneficial for the scientific community, all the proposed methodological modifications with the relevant statistical evaluations must be completed. Undoubtedly you also need to state a clear hypothesis of your study at the end of the Introduction. You must state your main observation that drove you to perform the study to prove it.
• Are the references appropriate?
You should update your references according to the most recent publications because only 3 out of 35 references are dated in 2022 and we have no information similar or different to your observations. This will give further strength in your discussion section and to more solid conclusions.
• Any additional comments on the tables and figures.
Some supplemental material should be incorporated in the submitted manuscript, especially the anthropometric and the physical examination results. Besides that, the new statistical evaluation might reveal the need either to update the tables and figures or to add new ones.
Comments on the Quality of English LanguageThe English could be improved to more clearly express the research performed and upgrade the quality of your presentation
